

# 1  Characterization of soil organic matter by near infrared

# 2  spectroscopy - determination of glomalin in different soils

**Jiří Zbíral[1], David Čižmár[1], Stanislav Malý[1] and Elena Obdržálková[1]**
[1] Central Institute for Supervising and Testing in Agriculture, Brno, Czech Republic
Correspondence to: J. Zbíral, (jiri.zbiral@ukzuz.cz)

## 8  Abstract

Determining and characterizing soil organic matter (SOM) cheaply and reliably can help to support
decisions concerning sustainable land management and climate policy. Glomalin, a glycoprotein
produced by arbuscular mycorrhizal fungi, was recommended as a promising indicator of SOM quality.
But extracting glomalin from and determining glomalin in soils using classical chemical methods is too
complicated and time consuming and therefore limits the use of this parameter in large scale surveys.
Near infrared spectroscopy (NIRS) is a very rapid, non-destructive analytical technique that can be used
to determine many constituents of soil organic matter.
Representative sets of 84 different soil samples from arable land and grasslands and 75 forest soils were
used to develop reliable NIRS calibration models for glomalin. One calibration model was developed for
samples with a low content of glomalin (arable land and grasslands), the second for soils with a high
content of glomalin (forest soils), and the third calibration model for all combined soil samples.
Calibrations were validated and optimized by leave-one-sample-out-cross-validation (LOSOCV) and by
the external validation using eight soil samples (arable land and grassland), and six soil samples (forest
soils) not included in the calibration models.
Two different calibration models were recommended. One model for arable and grassland soils and the
second for forest soils. No statistically significant differences were found between the reference and the



NIRS method for both calibration models. The parameters of the NIRS calibration model (RMSECV =
0,70 and R = 0,90 for soils from arable land and grasslands and RMSECV = 3,8 and R = 0,94 for forest
soils) proved that glomalin can be determined directly in air-dried soils by NIRS with adequate trueness
and precision.

## 1. Introduction

Glomalin, a glycoprotein produced by arbuscular mycorrhizal fungi, was discovered and partially
characterized in 1996 (Wright and Upadhyaya, 1996; Wright et al., 1996). It is assumed that the first
function of glomalin is to protect hyphae from water and nutritient loss. But glomalin is also one of the
factors that plays an important role in the formation and stabilization of soil aggregates (Wright and
Upadhyaya, 1998; Rillig, 2004; Purin and Rillig, 2007). Glomalin presence increases water retention,
nutrient cycling and reduces soil erosion. Glomalin also contributes to the improvement of soil porosity,
development of root systems, relevant soil enzyme activities and plant growth (Wang et al., 2015).
Glomalin contains approximately 37 % carbon and, in the soil environment, is characterized by
persistence ranging from several months to years (Rillig et al., 2001; Rillig, 2004). Therefore glomalin is
supposed to be an important part of the terrestrial carbon pool reducing atmospheric carbon dioxide levels.
The role of glomalin in ecosystems and the influence of land use on its content and stability was studied
by Treseder and Turner (2007) and Bedini et al. (2007), among others. It was found that glomalin can be
used as an effective indicator of soil quality (Fokom et al., 2012, Vasconcellos et al., 2013) and as one of
the criteria to define agricultural management strategies (Fokom et al, 2013). Glomalin was assumed to
be a sensitive indicator of soil carbon changes (Rillig et al., 2003).
Characterizing glomalin as a separate and unique fraction of soil organic matter is a complicated task
(Nichols, 2003; Nichols and Wright, 2005; Schindler et al., 2007). The link between glomalin and various
protein fractions in soil is not clearly defined. Co-extraction of non-glomalin proteins cannot be avoided
and glomalin-related soil protein (GRSP) was proposed as an operationally defined parameter correlating
with the ecosystem parameters of interest (Rillig, 2004). Although GRSP is only operationally defined
and influenced by the extraction procedure and the method of determination, it can be used as a parameter
relating closely to soil quality.



Glomalin is usually extracted from soils using 50 mM sodium citrate at pH 8 at 121°C in several one hour
cycles. Comparison of the efficacy of three different extractants was studied by Wright et al. (2006). The
authors found that sodium pyrophosphate or sodium borate were more effective at extracting glomalin
than sodium citrate. Rosier et al. (2006) showed that the extraction process cannot eliminate all non-
glomalin protein sources also determined by the Bradford assay (Bradford, 1976) because the Bradford
assay detects all peptides larger than 3 kDa. Janos et al. (2007) studied autoclaving duration and delayed
centrifugation, extraction ratio, spike recovery and denaturation of proteins during autoclaving. The
authors concluded that although glomalin is partially denatured by heat and pressure during extraction,
autoclaving is necessary to efficiently remove glomalin from soils. They suggest paying attention to the
extraction procedure (extraction volume, autoclaving time) and to the contact of soil with the extract after
extraction (they recommend immediate centrifugation). On the contrary to these results Lu et al. (2011)
found that for the total glomalin content there is no significance difference between centrifugation delays
of up to two hours. Pérez et al. (2012) compared two extractants with different extraction effectivity for
glomalin. They concluded that more aggressive extractants than citrate and extractants with a higher
concentration are more likely to remove some additional fractions of humic substances.
Interferences in the Bradford assay caused by co-extracted humic substances for forest soils were studied
by Jorge-Araújo et al. (2014). The authors concluded that the exact quantification of soil protein is
complicated by the positive interference of non-proteins and the negative interference in the Bradford
assay by co-extracted humic substances even though determination of GRSP can be useful for comparison
of soil protein in different soils or for studies of variations for a given soil due to season, climate or land
use.
Soil extraction followed by the Bradford assay determination has been studied also by Koide and Peoples
(2013). The authors found that even if the Bradford assay suffers from many technical difficulties
(quantification of non-glomalin soil proteins, interferences from co-extracted phenolic substances) the
method can effectively predict glomalin content in hot citrate soil extracts in mineral soils.
Near infrared reflectance spectroscopy (NIRS) is a very fast non-destructive and environmentally friendly
analytical technique. This method is mainly used to analyse food, feed, pharmaceuticals etc. but it has



proved to be a very effective method for the basic characterization of some soil constituents (Shepherd et
al., 2005; McCarty and Reeves, 2006; Cheng-Wen Chang and Laird, 2002; Jia et al., 2014), for the
prediction of some chemical and biological soil properties (Heinze et al., 2013), and for a rapid and cost-
effective quantification of some soil quality indices (Askari et al., 2015). Calibration equations reflect the
relationship between the constituents of the sample and the NIRS spectral information (Martens and Nas,
1989; Nas et al., 2002; Barnes et al., 1989; Stone, 1974; Williams and Norris, 2001). Central Institute for
Supervising and Testing in Agriculture (UKZUZ) has developed and optimized the NIRS method for
determining carbon and nitrogen in soils and prepared this method for international standardization in
ISO TC 190 Soil quality (ISO, 2014). It was assumed that more information, including information about
GRSP content, could be retrieved from the NIRS soil spectra.
We decided to focus our research mostly on these questions:
Can the measurement and calibration procedure described in the ISO standard for carbon and nitrogen
determination by NIRS also be applied for the determination of GRSP?
Is the reference method and the NIRS method applicable for the whole range of agriculture and forest
soils and contents of GRSP?

## 2.   Materials and methods

### 2.1. Soil samples

Soil samples from the UKZUZ regular monitoring plots, 68 on arable land and 24 on grasslands, were
used for the study (soil types: albeluvisol 6 samples, cambisol 26 samples, chernozem 8 samples, fluvisol
1 sample, gleysol 6 samples, haplic luvisol 25 samples, leptosol 3 samples, phaeozem 1 sample, planosol
6 samples, regosol 1 sample, technosol 1 sample). The soils covered a wide range of soils with different
content of organic matter (Table 1). Air-dried soil samples, fraction < 2 mm, were used for the study.
Eighty four samples were used for calibration and eight soil samples were used for external validation
(Table 2).



Eighty one forest soil samples from the F+H horizons represented the variability of forest ecosystems
across the Czech Republic (soil types: fluvisol 28 samples, cambisol 28 samples, albic podzol 20 samples,
and stagnosol 5 samples). Tree species forest composition and elevation, soil type, distance from cities
and distance from the forest edge were the main criteria of the sampling sites for the study (Borůvka et
al., 2015). Seventy five soil samples were used for the calibration model (Table 1) and six for external
validation (Table 3).

## 7 2.2. Reference method - Soil extraction and protein determination

The soil samples were extracted following the procedure described by Wright and Upadhyaya (1996).
Eight millilitres of a 50 mmol l$^{-1}$ sodium citrate solution (pH = 8.00) were added to 1.00 g of soil sample
in a 30ml plastic autoclavable tube and extracted by autoclaving at 121 °C and 1.4 kg cm$^{-2}$ for 60 min. A
steam sterilizer (75 S, H + P Labortechnik, Germany) was used for the extraction. Centrifugation at 3700
g for 15 min was started immediately after autoclaving. The supernatant was decanted and stored at 4 °C
until analysis but not more than three weeks. The soil was resuspended and the extraction step repeated
until only a light yellow colour of the supernatant was reached. Not more than 10 extraction cycles were
used.
The protein content in the extract was determined by the Bradford method (Bradford, 1976) using the
commercially available Bio–Rad Protein Assay kit (Bio–Rad Laboratories, USA). The analysis was
performed according to the instructions provided by the manufacturer. Precipitation after addition of the
Bradford reagent was observed for forest soils (not detected for soils with low content of organic matter)
and the method had to be optimized. Dilution of one volume of the soil extract with two volumes of the
extraction solution was finally found suitable for preventing precipitation. After this change of the
procedure the spectrophotometric determination was possible. The standard curve was prepared with
bovine serum albumin as a standard (0 – 300 µg ml$^{-1}$). Standard solutions or soil extracts diluted in
phosphate-buffered saline (10 µl) in three replicates were mixed with 200 µl of diluted dye reagent in
wells of a 96-well flat-bottomed microplates using a shaker (30 s, 600 min-1). The mixture was incubated
for 15 min. The absorbance was measured at 595 nm on a microplate reader (Versamax, Molecular



Devices, USA). The dye reagent was prepared by diluting one part Dye Reagent Concentrate with four
parts water and filtering through a Whatman #1 filter.

## 2.3. NIRS determination

The NIRS spectra were recorded on a Nicolet Antaris II (Thermo Fisher Scientific, U.S.A.). The
reflectance spectra were measured from 1000 to 2 500 nm, resolution 0,5 nm. The soil samples were
transferred to the sample compressed cells with 3cm diameter and the surface was levelled. The spectra
of the samples were scanned in 120 scans under continuous sample rotation. Windows of the sample cups
were carefully cleaned by a gentle stream of  compressed air between the individual measurements. The
spectra were processed using TQ Analyst 8 instrument software (Thermo Electron Corp., USA). The
spectra were transformed in two ways:
(1) using the Savitzky-Golay algorithm with a 3rd order polynomial (Savitzky and Golay, 1964) and
smoothed to reduce baseline variations and to enhance spectral features (Reeves et al.,2002)
(2) using a standard normal variate (SNV) to correct a light scattering for different particle sizes.
Principal component analysis (PCA) was then performed to check the spectral homogeneity of the dataset.

## 3.  Results and Discussion

## 3.1. Calibration

The spectra and the results of the GRSP content determined by the reference method were used to
calculate the NIRS calibration model. A scatter plot of reference values and NIRS predicted values for
arable and grassland soils is given in Fig. 1 and for forest soils in Fig. 2.
Two calibration sample sets (84 soil samples from arable land and grassland and 75 forest soil samples)
were selected to gain an evenly distributed coverage of the basic soil properties (Table 1) and most of the
possible spectral variability. Calibrations were performed by partial least square (PLS) regression. Leave-
one-sample-out-cross-validation (LOSOCV) was used to determine the optimum number of PLS





components required to calibrate the models and then calculate the predicted values of the calibration
subset in order to assess the robustness of the models. Eight PLS components were found to be an
optimum for our calibration model. One sample was left-out from the calibration set, and a model was
built with the remaining samples. The left-out sample was predicted by this model, and the whole
procedure was repeated by leaving out each sample in the calibration set (ISO, 2014; Centner et al., 1998).
The residuals of cross-validation predictions were pooled to calculate the root mean square error of cross
validation (RMSECV). The RMSECV were calculated as:
$$RMSECV = \sqrt{\frac{\sum_{i=1}^{n_C}(\hat{y}_{ci}-y_{ci})^2}{n_c}} \qquad (1)$$
where $n_C$ is the number of samples in calibration set, $y_{ci}$ is the reference measurement value of sample $i$,
and $\hat{y}_{ci}$   is the estimated value for sample $i$ by the model constructed when the sample $i$ is left out;
The final calibration model was chosen according to the global lowest RMSECV = 0,70 and R = 0,90 for
soils from arable land and grassland and RMSECV = 3,8 and R = 0,94 for forest soils. A calibration model
covering all soils can be also derived by the same procedure and the results are acceptable but the spectral
properties of forest soils and agriculture soils were substantially different (Fig. 3) and therefore we finally
decided to use two separate calibration models - one for soils from arable land and grassland with the
content of glomalin up to 12  mg g$^{-1}$ and the second for forest soils with the content of glomalin up to 60
mg g$^{-1}$.
**3.2. Validation of the calibration models**
The prediction ability of both calibration models was tested on independent sample sets (eight different
soil samples for arable land and grasslands and six samples for forest soils) by external validation. The
main characterization of the samples used for external validation and the results of the estimation of
glomalin content are given in Table 2 and Table 3. The content of GRSP was determined using a reference
method and NIRS in triplicate. The results were compared using R 3.0.2 software by paired t-test. The
analysis did not show any statistically significant difference between the reference method and the NIRS
method (p=0,55 for arable soils and grasslands, p=0,54 for forest soils).



## 3.3. Routine measurement

In a routine measurement, a principal component analysis (PCA) and a cluster analysis are applied to prove that the developed calibration model is valid for each individual soil sample. Samples that do not fulfil the statistical criteria are analysed by a reference method and the results are used to improve the calibration model. Other soil parameters such as the content of oxidizable carbon (Cox), total carbon and total nitrogen are determined simultaneously from the same NIRS measurement (ISO, 2014).

## 4. CONCLUSIONS

NIRS proved to be a very powerful technique to reliably and quickly determine GRSP. The method can substitute the relatively difficult and laborious determination of GRSP in soils by high-pressure extraction followed by Bradford protein determination. Our results support the results of many authors who used NIRS to determine a wide range of soil properties with this method (e.g. Peltre et al., 2011; Heinze et al., 2013). The NIRS method has been tested in a large scale in the national soil testing survey where, among others, the determination of the GRSP/Cox ratio was calculated as one of the possible markers of soil organic matter (SOM) quality. Our future work will focus on the improvement of the interpretation of the NIRS results as well as widening the scope of the NIRS calibration for other SOM quality markers.





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





1    Table 1. Physico-chemical soil properties of samples used for calibration (arable soils and grasslands,

2    84 samples; forest soils, 75 samples)

| | Arable and grassland soils | | | | Forest soils | | | |
|---|---|---|---|---|---|---|---|---|
| | pH | Cox (%) | CEC (me kg-1) | GRSP (mg g-1) | pH | Cox (%) | CEC (me kg-1) | GRSP (mg g-1) |
| Minimum | 3.6 | 1.1 | 91 | 1.1 | 3.3 | 3.7 | 60 | 6.6 |
| 1st quartile | 5.0 | 1.5 | 154 | 2.4 | 3.6 | 7.8 | 289 | 13.2 |
| Median | 5.6 | 1.9 | 199 | 3.2 | 4.0 | 13.3 | 406 | 17.8 |
| Mean | 5.6 | 2.2 | 210 | 3.6 | 4.2 | 16.2 | 457 | 21.0 |
| 3rd quartile | 6.2 | 2.5 | 243 | 4.6 | 4.5 | 25.7 | 610 | 25.9 |
| Maximum | 7.5 | 6.1 | 528 | 10.5 | 6.6 | 35.3 | 1100 | 55.1 |

3    CEC – cation exchange capacity,  Cox – oxidisable carbon, GRSP – content of glomalin



Table 2. Physico-chemical soil properties and the content of glomalin in samples used for external
validation – arable land and grasslands

|        | pH  | Cox (%) | CEC (me kg⁻¹) | GRSP (mg g⁻¹) | NIRS (mg g⁻¹) |
|--------|-----|---------|---------------|---------------|---------------|
| Soil 1 | 6.8 | 1.4     | 201           | 1.9           | 1.9           |
| Soil 2 | 7.3 | 1.9     | 215           | 3.9           | 4.2           |
| Soil 3 | 5.9 | 2.0     | 167           | 3.5           | 3.2           |
| Soil 4 | 6.7 | 3.0     | 317           | 3.7           | 3.7           |
| Soil 5 | 5.7 | 1.7     | 200           | 2.5           | 3.3           |
| Soil 6 | 6.6 | 1.5     | 143           | 2.8           | 3.1           |
| Soil 7 | 5.2 | 1.9     | 163           | 3.1           | 2.8           |
| Soil 8 | 6.1 | 1.8     | 171           | 2.9           | 2.9           |

CEC – cation exchange capacity, Cox – oxidisable carbon, GRSP – content of glomalin determined by
high-pressure extraction and Bradford assay, NIRS – content of glomalin determined by NIRS method



Table 3. Physico-chemical soil properties and the content of glomalin in samples used for external
validation – forest soils

|  | pH | Cox (%) | Ctot (%) | GRSP (mg g$^{-1}$) | NIRS (mg g$^{-1}$) |
|---|---|---|---|---|---|
| Soil 1 | 4.3 | 10.2 | 11.0 | 13.7 | 12.3 |
| Soil 2 | 3.9 | 15.6 | 15.3 | 19.4 | 16.8 |
| Soil 3 | 5.3 | 26.3 | 27.7 | 25.8 | 23.1 |
| Soil 4 | 4.7 | 23.8 | 25.8 | 31.3 | 32.1 |
| Soil 5 | 5.7 | 41.9 | 42.4 | 32.8 | 33.0 |
| Soil 6 | 4.4 | 34.4 | 36.2 | 35.9 | 38.3 |

Cox – oxidisable carbon, Ctot – total carbon, GRSP – content of glomaline determined by high-pressure
extraction and Bradford assay, NIRS – content of glomalin determined by NIRS method.


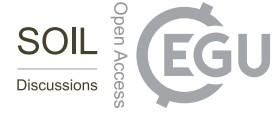

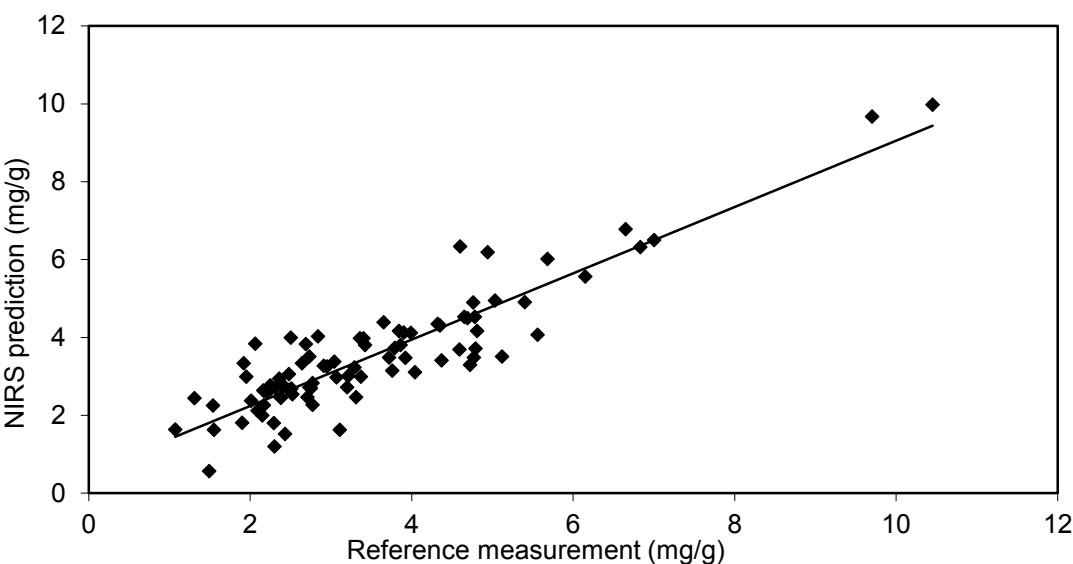

2    Figure 1. Regression curve for arable soils and grasslands



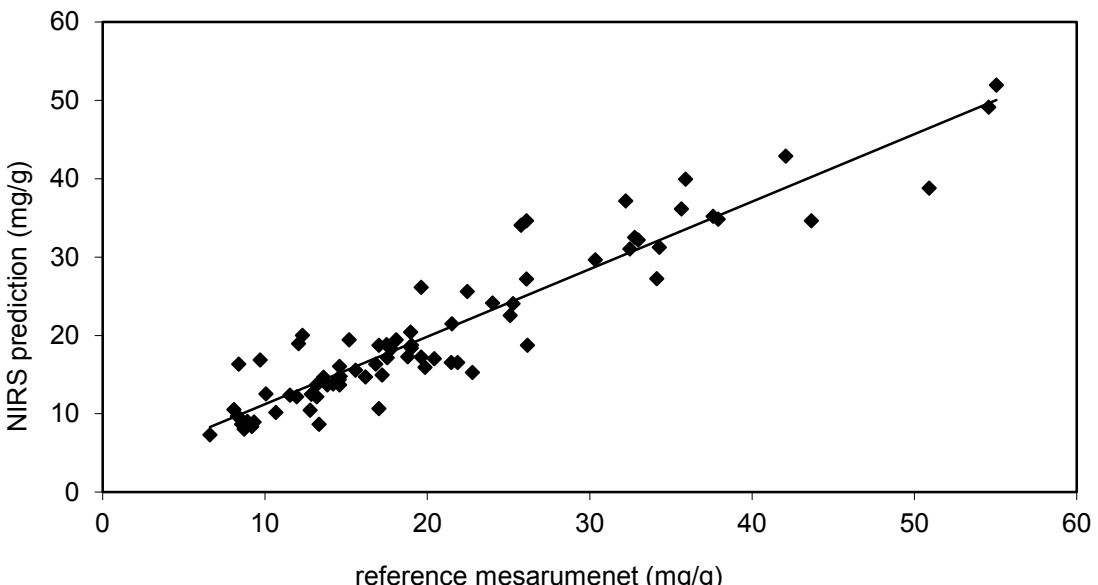

1                                                                             Figure 2.

2     Regression curve for organic horizon of forest soils.





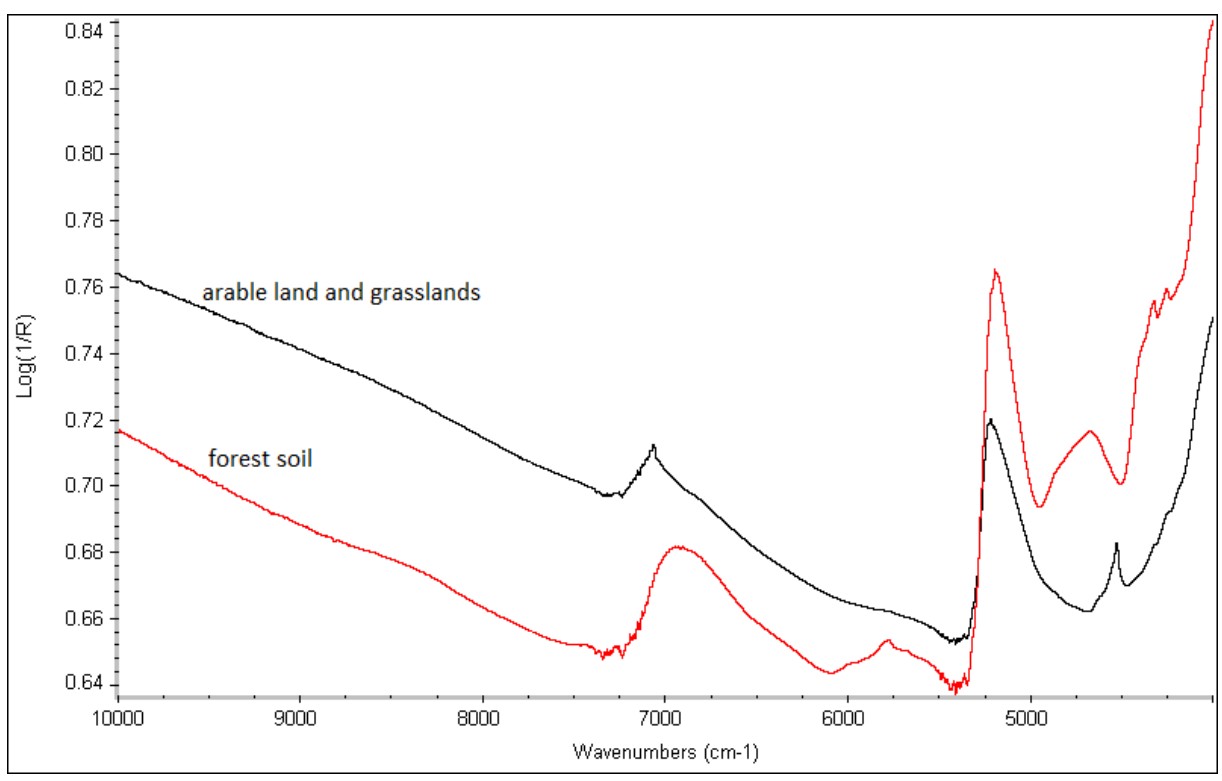

Figure 3. Example of typical NIRS spectra for forest soils and for arable and grassland soils.

