# Peer review of "Characterization of soil organic matter by near infrared 1 spectroscopy - determination of glomalin in different soils"

_SOIL, 2016_

## Referee Comment (RC1) · Anonymous Referee #1 · 11 Apr 2016

Commonly, glomalin-extraction is associated with the co-extraction of humic acids. Although the latter are not well defined due to the fact that it is an operational classification rather than a chemical, they contain a considerable amount of carbohydrates and proteins. Up to now I have not seen convincing evidence that glomalin is not a part of the proteinaceous fraction of humic acids. Considering further that humic acids are thought to fulfill exactly the same advantages with respect to soil fertility, the question arises what exactly is the difference? In the present paper, too, the glomalin was extracted with sodium citrate solution and the protein content was determined with the Bradford method. It is still not clear to me, how the obtained proteins are distinguishable from common soil proteins or cell wall residues which accumulate during

soil organic matter formation? Thus, in my opinion the authors have determined the amount of soil proteins which canIt is also not clear to me which specific parameter of the NIRS method is used for the identification of GRSP. The author may have developed a good means for determining the content of proteinaeous material which is extractable with sodium citrate but there is no prove that this is really glomalin. Taking this in account, I cannot see the new aspect of this work and I cannot recommend this paper for publication.

---

## Referee Comment (RC2) · Anonymous Referee #2 · 15 Apr 2016

General comments:

The paper presents a new approach for determining the presence of arbuscular mycor-rhizal fungi in soils, through the detection of the glomalin content. They aim at detecting glomalin by NIR spectroscopy. Although in general well written and fairly structured my main comment is that the authors do not show convincing evidence that what they are determining actually is the glomalin content (or GRSP fraction).

I was surprised that the authors chose a NIRS signal as a proxy, as it is well established in literature that NIRS of soils give very broad, superimposed vibration peaks generally not suitable for assessing soil organic matter components in detail. In order to see individual peaks that can be related to the vibrations of specific chemical bonds, MIRS

are usually required. NIRS is most successful in assessing total organic carbon.

Hence, when looking at the rather scarce data provided regarding the samples, the GRSP seems to correlate strongly with the total or oxidisable C present and figures 1 and 2 indicate quite some scatter. I am therefore not convinced that the NIRS successfully determines GRSP: it may actually determine carbon content, which may happen to correlate with GRSP. Hence, I would like to see some statistical analysis to establish if the two are correlated, and the same graphs as in figure 1 and 2, but with the NIRS prediction for GRSP plotted against the total organic carbon content.

A second major comment is that – although section 3 is labeled "Results and Discussion", I do not see a lot of discussion in the text. The section is almost exclusively a description of the results.

Specific comments:

- In the abstract and the introduction, glomalin is repeatedly described as a promising indicator of SOM quality (or soil carbon changes, page 2 - line 20). Given the substantial difficulty in measuring glomalin, it is not very suitable as an indicator: many other proxies for SOM quality/dynamics exist, which are much easier to obtain. Hence, the authors should clearly indicate the additional value of globalin content for e.g. large scale surveys (page 1 -line 13) as compared to existing indicators.

- The rest of the introduction is mainly a methodological state of the art, towards the procedures to be followed to extract and process the GRSP. For a reader not too familiar with the protocol, it is quite confusing. Many problems are listed, but in the end it is concluded that the method is nevertheless valid, yet no strong arguments for that point are provided.

- For NIRS: the introduction is very general. Some critical reflections about the possibilities and limitations of the technique are in order, either in the introduction or in the conclusion

[Figure]

- Introduction and materials and methods: There are many types of NIRS, which one did you use (DRIFT?)

- Section 2.1: mention which soil classification system you use and use correct terms (e.g. Fluvisol, Cambisol etc is written with a capital in WRB).

- Section 2.3: specify the type of NIR (diffuse reflected? Fourier transformed? Etc). Much more interesting than the type of software is the type of statistical analysis it performs.

- Section 3.1: specify further the kind of model that was used and how optimalisations were done. Were all wavelengths included in the final model?

- Section 3.3: Not clear, description of the method rather than a result.

- Table 1, 2 and 3: why are different soil properties listed in the tables? Sometimes CeS is measured, sometimes Cox and sometimes both Cox and Ctot. I do not see a rationale in it. Also, these results are not described nor discussed.

- Figure 1: two observations clearly have a higher value for both the measurements and the prediction, with a big gap between them and the rest. How good is your model if you repeat it without these two observations? Also, indicate what the line depicts. A regression plotted between the two? Indicate an R2 value if this is the case. Idem for figure 2.

- Figure 3: why the difference in baseline? 1 sample or an average? Specify

Technical:

- The UKZUZ is a national institute, this should be specified more clearly - Page 4, lines 19-25: structure is confusing as first it seems that there should be 92 samples in table 1, etc. - In the results section, you often mention using "a reference method" – do you mean the one specified in the materials and methods section? Then clearly indicate so. - Many typo's regarding units – consistently leave a space between the value and

the unit and do not leave spaces in values (e.g. page 6 - line 5; 2500 in stead of 2 500

---

## Author Comment (AC1) · 15 Apr 2016

The problems associated with the definition of glomalin as a unique protein fraction has been mentioned in our paper (1). It was not our intention to discuss these issues and it was clearly stated in the paper (2). Our intention was to prepare a tool for an easy and cheap determination of GPRS and the goal was achieved. We think that the discussion concerning GPRS is often based on very few data due to serious difficulties with the analytical methodology. Both – extraction step and Bradford determination are not easy, not cheap and not fast. Therefore we suppose that application of NIRS could be very useful to improve discussion after more data are collected. Despite of co-extraction of various proteins and other compounds along with glomalin, GPRS

extracted by means of the citrate buffer or citric acid is still used as an indicator of soil quality related to the presence of AMF (e.g. DOI 10.1007/s00572-014-0572-9). Analysis of extracted proteins based on the Bradford reagent has recently been shown to be linked to the content of glomalin (http://dx.doi.org/10.1016/j.apsoil.2012.09.015f). We wanted to provide an effective analytical tool which could facilitate determination of GRSP (and our goal was clearly stated in the paper). The anonymous reviewer 1 commented problems that were not in the scope of our work.

[1]"Characterizing glomalin as a separate and unique fraction of soil organic matter is a complicated task (Nichols, 2003; Nichols and Wright, 2005; Schindler et al., 2007). The link between glomalin and various protein fractions in soil is not clearly defined. Co-extraction of non-glomalin proteins cannot be avoided and glomalin-related soil protein (GRSP) was proposed as an operationally defined parameter correlating with the ecosystem parameters of interest (Rillig, 2004). Although GRSP is only operationally defined and influenced by the extraction procedure and the method of determination, it can be used as a parameter relating closely to soil quality.

[2] "Central Institute for Supervising and Testing in Agriculture (UKZUZ) has developed and optimized the NIRS method for determining carbon and nitrogen in soils and prepared this method for international standardization in ISO TC 190 Soil quality (ISO, 2014). It was assumed that more information, including information about GRSP content, could be retrieved from the NIRS soil spectra. We decided to focus our research mostly on these questions: Can the measurement and calibration procedure described in the ISO standard for carbon and nitrogen determination by NIRS also be applied for the determination of GRSP? Is the reference method and the NIRS method applicable for the whole range of agriculture and forest soils and contents of GRSP?

---

## Author Comment (AC2) · 26 Apr 2016

Reply to RC 2 comments A. General remarks 1) Suitability of NIRS It is true that soils have very broad peaks generally not suitable for assessing individual soil organic matter components. Soils generally have very similar reflectance spectra in the 1100 to 2500 nm range (FIGURE 3 shows that differences can be seen between soils from arable land and forest soils). But even if the absorption peaks for soils in the near-infrared region are difficult to assign to specific chemical components, the spectra still have complex information that can be used for analytical purposes. ISO 17184 clearly shows that even if the individual peaks cannot be assigned to any individual compound the spectral information reflects organic matter content in soil. Glomalin is not defined as an individual compound but as an operational parameter GRSP. The situation is exactly the same as if for example fibre is determined by NIRS in feed. 2) GRSP and Cox correlation As Cox is (GRSP+other SOM components) there is really a correlation between the parameters. But this correlation is not so strong that we could say that GRSP is not bringing any new information. We suppose that mainly the ratio GRSP/Cox could be probably valuable qualitative indicator of SOM. But it was not intended to discuss this issue in our paper. Only for your information we add an Excel sheet with regression results as you proposed. It can be seen that we cannot say that GRSP could be easily calculated from Cox. The discussion is maybe for other paper and it was not in the scope of our paper. We intended mainly to propose a new and effective analytical tool for simultaneous determination of Cox and GRPS (and maybe more SOM parameters in future). 3) Results and discussion The discussion part was improved in the section 3.

B. Specific comments 1) In the abstract and the introduction, glomalin is repeatedly described as a promising indicator of SOM quality (or soil carbon changes, page 2 - line 20). Given the substantial difficulty in measuring glomalin, it is not very suitable as an indicator: many other proxies for SOM quality/dynamics exist, which are much easier to obtain. Hence, the authors should clearly indicate the additional value of globalin content for e.g. large scale surveys (page 1 -line 13) as compared to existing indicators. In the references we found that authors suppose that GRSP can be a valuable SOM quality indicator and these citations were used in the Introduction part. In fact it was not our task (at least not in this paper) to show how useful or not GRSP is. However, we assume that the value of this parameter as an indicator of soil quality can rely in its contribution to aggregate stability (and related physical soil properties) which results from slow decomposition of GRPS. Most analytical methods for estimation of various fractions of SOM are rather focused on labile fractions which can be used as early indicators of changes in dynamics of SOM (e.g. water extractable C, light fraction, particulate SOM).We showed that there is a way how to determine this "difficult to measure indicator" in a substantially easier way. And this may facilitate further research about usefulness of this indicator. We fully agree that GRSP cannot be used instead of other SOM indicators but we suppose that it can be used as an additional parameter. In the large scale surveys any indicator that can be determined by NIRS simultaneously with e.g. Cox or Ctot is extremely valuable because it is very cheap. The possibility of this simultaneous determination was be emphasized in the revised text. 2) The rest of the introduction is mainly a methodological state of the art, towards the procedures to be followed to extract and process the GRSP. For a reader not too familiar with the protocol, it is quite confusing. Many problems are listed, but in the end it is concluded that the method is nevertheless valid, yet no strong arguments for that point are provided. We wanted to show that there was some effort to improve extraction and/or measurement step of the former analytical method for GRSP determination but most authors finally agreed that the method published by Wright and Upadhyaya is acceptable. And this method was used in our work as a reference method.

(more points aggregated) For NIRS: the introduction is very general. Some critical reflections about the possibilities and limitations of the technique are in order, either in the introduction or in the conclusion - Introduction and materials and methods: There are many types of NIRS, which one did you use (DRIFT?) Section 2.3: specify the type of NIR (diffuse reflected? Fourier transformed? Etc).Much more interesting than the type of software is the type of statistical analysis it performs. Specify further the kind of model that was used and how optimalisations were done. Were all wavelengths included in the final model. Accepted. The text will be improved in this way. FT diffuse reflectance was used. 4) Mention which soil classification system you use and use correct terms (e.g. Fluvisol, Cambisol etc is written with a capital in WRB) Accepted, corrected 5) Section 3.3: Not clear, description of the method rather than a result. In fact, the main result of our work was a new analytical method and this method is described in this section. We can put it in the "conclusion" but we suppose that as a final result of our work it should remain in the Result part. 6) Table 1, 2 and 3: why are different soil properties listed in the tables? Sometimes CeS is measured, sometimes Cox and sometimes both Cox and Ctot. I do not see a rationale in it. Also, these results are not described nor discussed. The available data for arable land and grasslands and for forest soils were different, but it is really confusing therefore we adjusted the tables to have only the same parameters. We did not discuss the parameters. They were shown only to support part 2.1 where we stated that the sample set covered wide range of different soils. The text was corrected.

7) Two observations clearly have a higher value for both the measurements and the prediction, with a big gap between them and the rest. How good is your model if you repeat it without these two observations? Also, indicate what the line depicts. A regression plotted between the two? Indicate an R2 value if this is the case. Idem for figure 2.

Accepted and corrected. Calculation was done and the results are discussed in the Part 3. 8) Why the difference in baseline? 1 sample or an average? Specify We do not fully understand the first part of the comment. In Figure 3 there are examples of a typical soil spectra for a soil from arable land and a forest soil. It is an example (1 soil). Description of the Figure was changed to be clear.

C. Technical comments The UKZUZ is a national institute, this should be specified more clearly – We do not fully understand this comment. Page 4, lines 19-25: structure is confusing as first it seems that there should be 92 samples in table 1, etc. The total number of soils from arable land and grasslands was 92, Out of them 84 (Table 1) was chosen for calibration of the NIRS instrument and the remaining 8 (Table 2) was used for external validation. The total number of forest soils was 81. 75 for calibration of NIRS (Table 1) and 6 for external validation (Table 3).

- In the results section, you often mention using "a reference method" – do you mean the one specified in the materials and methods section? Then clearly indicate so. – Yes, it is the method described in the part 2.2 "Reference method. . . . . .." The text was changed to avoid confusion.

Many typo's regarding units – consistently leave a space between the value and the unit and do not leave spaces in values (e.g. page 6 - line 5; 2500 in stead of 2 500 The space was deleted (we have found only this one). We cannot see any space that should be added between number and unit. Only in page.5, line 10, but there is 30ml as an adjective and we are not sure if a space is possible in this case.

Please also note the supplement to this comment:
http://www.soil-discuss.net/soil-2016-9/soil-2016-9-AC2-supplement.pdf

[Figure]

Relationship between Cox and GRSP.

**Fig. 1.** reply to RC 2 comments - Cox vs. GRSP

**Supplement:**

[revised manuscript text omitted]

84 samples; forest soils, 75 samples)

| | Arable and grassland soils | | | Forest soils | | |
|---|---|---|---|---|---|---|
| | pH | Cox (%) | GRSP (mg g-1) | pH | Cox (%) | GRSP (mg g-1) |
| Minimum | 3.6 | 1.1 | 1.1 | 3.3 | 3.7 | 6.6 |
| 1st quartile | 5.0 | 1.5 | 2.4 | 3.6 | 7.8 | 13.2 |
| Median | 5.6 | 1.9 | 3.2 | 4.0 | 13.3 | 17.8 |
| Mean | 5.6 | 2.2 | 3.6 | 4.2 | 16.2 | 21.0 |
| 3rd quartile | 6.2 | 2.5 | 4.6 | 4.5 | 25.7 | 25.9 |
| Maximum | 7.5 | 6.1 | 10.5 | 6.6 | 35.3 | 55.1 |

CEC – cation exchange capacity,  Cox – oxidisable carbon, GRSP – content of glomalin

Naformátována tabulka

Table 2. Physico-chemical soil properties and the content of  GRSP in samples used for external validation – arable land and grasslands

| | pH | Cox (%) | GRSP (mg g$^{-1}$) | NIRS (mg g$^{-1}$)[a] | NIRS (mg g$^{-1}$)[b] |
|---|---|---|---|---|---|
| Soil 1 | 6.8 | 1.4 | 1.9 | 1.9 | 2.1 |
| Soil 2 | 7.3 | 1.9 | 3.9 | 4.2 | 4.2 |
| Soil 3 | 5.9 | 2.0 | 3.5 | 3.2 | 2.9 |
| Soil 4 | 6.7 | 3.0 | 3.7 | 3.7 | 4.4 |
| Soil 5 | 5.7 | 1.7 | 2.5 | 3.3 | 3.0 |
| Soil 6 | 6.6 | 1.5 | 2.8 | 3.1 | 3.0 |
| Soil 7 | 5.2 | 1.9 | 3.1 | 2.8 | 2.7 |
| Soil 8 | 6.1 | 1.8 | 2.9 | 2.9 | 3.2 |

CEC – cation exchange capacity, Cox – oxidisable carbon, GRSP – content of glomalin determined by high-pressure extraction and Bradford assay; NIRS - content of GRSP determined by NIRS method (NIRS[a] calibration model for all samples, content of GRSP < 12 mg g$^{-1}$; NIRS[b] calibration model without two high samples, content of GRSP < 8 mg g$^{-1}$).

Table 3. Physico-chemical soil properties and the content of  GRSP in samples used for external validation – forest soils

|        | pH  | Cox (%) | GRSP (mg g$^{-1}$) | NIRS (mg g$^{-1}$) |
|--------|-----|---------|--------------------|--------------------|
| Soil 1 | 4.3 | 10.2    | 13.7               | 12.3               |
| Soil 2 | 3.9 | 15.6    | 19.4               | 16.8               |
| Soil 3 | 5.3 | 26.3    | 25.8               | 23.1               |
| Soil 4 | 4.7 | 23.8    | 31.3               | 32.1               |
| Soil 5 | 5.7 | 41.9    | 32.8               | 33.0               |
| Soil 6 | 4.4 | 34.4    | 35.9               | 38.3               |

Cox – oxidisable carbon, Ctot – total carbon, GRSP – content of glomaline determined by high-pressure extraction and Bradford assay, NIRS – content of glomalin determined by NIRS method.

Naformátována tabulka

[Figure]

Figure 1. Regression curve for arable soils and grasslands. R = 0.90.

[Figure]

Figure 2. Scatter plot and rRegression curve for organic horizon of forest soils. R = 0,94.

[Figure]

Figure 3. Example of  NIRS spectra for a typical forest soil and  an arable and grasslands soil.